# STOCHASTIC SECURITY: ADVERSARIAL DEFENSE USING LONG-RUN DYNAMICS OF ENERGY-BASED MODELS

**Mitch Hill** [*]
Department of Statistics and Data Science
University of Central Florida
mitchell.hill@ucf.edu

**Jonathan Mitchell** [* †] **& Song-Chun Zhu** [† ‡]
Department of Computer Science[†]
Department of Statistics[‡]
University of California, Los Angeles
jcmitchell@ucla.edu
sczhu@stat.ucla.edu

## ABSTRACT

The vulnerability of deep networks to adversarial attacks is a central problem for deep learning from the perspective of both cognition and security. The current most successful defense method is to train a classifier using adversarial images created during learning. Another defense approach involves transformation or purification of the original input to remove adversarial signals before the image is classified. We focus on defending naturally-trained classifiers using Markov Chain Monte Carlo (MCMC) sampling with an Energy-Based Model (EBM) for adversarial purification. In contrast to adversarial training, our approach is intended to secure highly vulnerable pre-existing classifiers. To our knowledge, no prior defensive transformation is capable of securing naturally-trained classifiers, and our method is the first to validate a post-training defense approach that is distinct from current successful defenses which modify classifier training.

The memoryless behavior of long-run MCMC sampling will eventually remove adversarial signals, while metastable behavior preserves consistent appearance of MCMC samples after many steps to allow accurate long-run prediction. Balancing these factors can lead to effective purification and robust classification. We evaluate adversarial defense with an EBM using the strongest known attacks against purification. Our contributions are 1) an improved method for training EBM's with realistic long-run MCMC samples for effective purification, 2) an Expectation-Over-Transformation (EOT) defense that resolves ambiguities for evaluating stochastic defenses and from which the EOT attack naturally follows, and 3) state-of-the-art adversarial defense for naturally-trained classifiers and competitive defense compared to adversarial training on CIFAR-10, SVHN, and CIFAR-100. Our code and pre-trained models are available at https://github.com/point0bar1/ebm-defense.

## 1 MOTIVATION AND CONTRIBUTIONS

Deep neural networks are highly sensitive to small input perturbations. This sensitivity can be exploited to create adversarial examples that undermine robustness by causing trained networks to produce defective results from input changes that are imperceptible to the human eye (Goodfellow et al., 2015). The adversarial scenarios studied in this paper are primarily untargeted white-box attacks on image classification networks. White-box attacks have full access to the classifier (in particular, to classifier gradients) and are the strongest attacks against the majority of defenses.

Many whitebox methods have been introduced to create adversarial examples. Strong iterative attacks such as Projected Gradient Descent (PGD) (Madry et al., 2018) can reduce the accuracy of a naturally-trained classifier to virtually 0. Currently the most robust form of adversarial defense is to train a classifier on adversarial samples in a procedure known as *adversarial training* (AT) (Madry

---

[*]Equal Contribution

et al., 2018). Another defense strategy, which we will refer to as *adversarial preprocessing* (AP), uses defensive transformations to purify an image and remove or nullify adversarial signals before classification (Song et al. (2018); Guo et al. (2018); Yang et al. (2019), and others). AP is an attractive strategy compared to AT because it has the potential to secure vulnerable pre-existing classifiers. Defending naturally-trained classifiers is the central focus of this work.

Athalye et al. (2018) revealed that many preprocessing defenses can be overcome with minor adjustments to the standard PGD attack. Both stochastic behavior from preprocessing and the computational difficulty of end-to-end backpropagation can be circumvented to attack the classifier through the defensive transformation. In this paper we carefully address Athalye et al. (2018) to evaluate AP with an EBM using attacks with the greatest known effectiveness and efficiency.

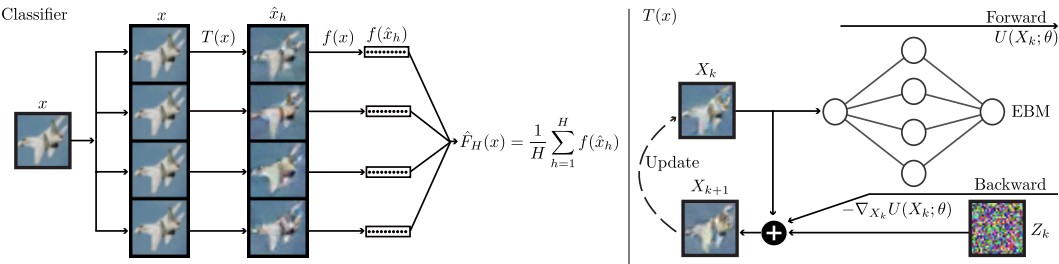

Figure 1: *Left:* Visualization of calculating our stochastic logits $\hat{F}_H(x)$ from (5). The input image $x$ is replicated $H$ times and parallel Langevin updates with a ConvNet EBM are performed on each replicate to generate $\{\hat{x}_h\}_{h=1}^{H}$. Purified samples are sent in parallel to our naturally-trained classifier network $f(x)$ and the resulting logits $\{f(\hat{x}_h)\}_{h=1}^{H}$ are averaged to produce $\hat{F}_H(x)$. The logits $\hat{F}_H(x)$ give an approximation of our true classifier logits $F(x)$ in (4) that can be made arbitrarily precise by increasing $H$. *Right:* Graphical diagram of the Langevin dynamics (3) that we use for $T(x)$. Images are iteratively updated with a gradient from a naturally-trained EBM (1) and Gaussian noise $Z_k$.

Langevin sampling using an EBM with a ConvNet potential (Xie et al., 2016) has recently emerged as a method for AP (Du & Mordatch, 2019; Grathwohl et al., 2020). However, the proposed defenses are not competitive with AT (see Table 1 and Croce & Hein (2020)). In the present work we demonstrate that EBM defense of a naturally-trained classifier can be stronger than standard AT (Madry et al., 2018) and competitive with state-of-the-art AT (Zhang et al., 2019; Carmon et al., 2019).

Our defense tools are a classifier trained with labeled natural images and an EBM trained with unlabeled natural images. For prediction, we perform Langevin sampling with the EBM and send the sampled images to the naturally-trained classifier. An intuitive visualization of our defense method is shown in Figure 1. Langevin chains constitute a memoryless trajectory that removes adversarial signals, while metastable sampling behaviors preserve image classes over long-run trajectories. Balancing these two factors leads to effective adversarial defense. Our main contributions are:

- A simple but effective adjustment to improve the convergent learning procedure from Nijkamp et al. (2020). Our adjustment enables realistic long-run sampling with EBMs learned from complex datasets such as CIFAR-10.
- An Expectation-Over-Transformation (EOT) defense that prevents the possibility of a stochastic defense breaking due to random variation in prediction instead of an adversarial signal. The EOT attack (Athalye et al., 2018) naturally follows from the EOT defense.
- Experiments showing state-of-the-art defense for naturally-trained classifiers and competitive defense compared to state-of-the-art AT.

## 2 IMPROVED CONVERGENT LEARNING OF ENERGY-BASED MODELS

The Energy-Based Model introduced in (Xie et al., 2016) is a Gibbs-Boltzmann density

$$p(x;\theta) = \frac{1}{Z(\theta)} \exp\{-U(x;\theta)\} \tag{1}$$

where $x \in \mathbb{R}^D$ is an image signal, $U(x;\theta)$ is a ConvNet with weights $\theta$ and scalar output, and $Z(\theta) = \int_{\mathcal{X}} \exp\{-U(x;\theta)\}dx$ is the intractable normalizing constant. Given i.i.d. samples from a data distribution $q(x)$, one can learn a parameter $\theta^*$ such that $p(x;\theta^*) \approx q(x)$ by minimizing the expected negative log-likelihood $\mathcal{L}(\theta) = E_q[-\log p(X;\theta)]$ of the data samples. Network weights $\theta$ are updated using the loss gradient

$$\nabla \mathcal{L}(\theta) \approx \frac{1}{n} \sum_{i=1}^{n} \nabla_\theta U(X_i^+;\theta) - \frac{1}{m} \sum_{i=1}^{m} \nabla_\theta U(X_i^-;\theta) \tag{2}$$

where $\{X_i^+\}_{i=1}^n$ are a batch of training images and $\{X_i^-\}_{i=1}^m$ are i.i.d. samples from $p(x;\theta)$ obtained via MCMC. Iterative application of the Langevin update

$$X_{k+1} = X_k - \frac{\tau^2}{2} \nabla_{X_k} U(X_k;\theta) + \tau Z_k, \tag{3}$$

where $Z_k \sim \mathrm{N}(0, I_D)$ and $\tau > 0$ is the step size parameter, is used to obtain the samples $\{X_i^-\}_{i=1}^m$.

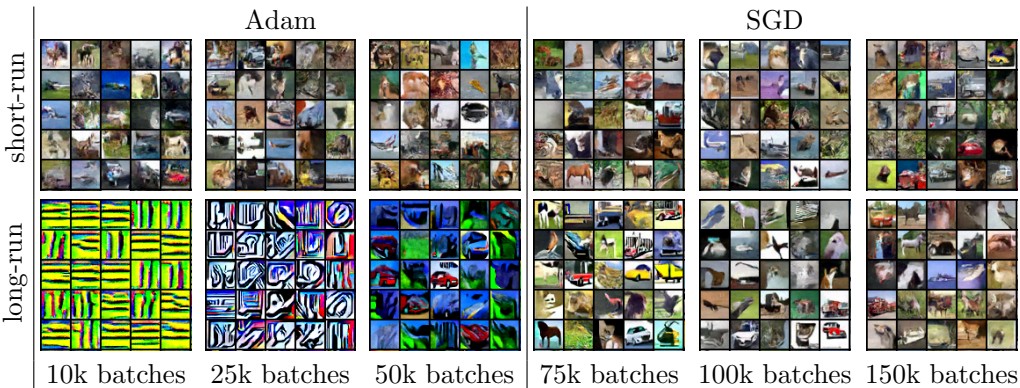

Figure 2: Comparison of long-run and short-run samples over model updates for our improved method of convergent learning. The model is updated in a non-convergent learning phase with the Adam optimizer for the first 50,000 batches. The majority of short-run synthesis realism is learned during this phase, but the long-run samples are very unrealistic. The second learning phase uses SGD with a low learning rate. Short-run synthesis changes very little, but the long-run distribution gradually aligns with the short-run distribution.

Nijkamp et al. (2020) reveal that EBM learning heavily gravitates towards an unexpected outcome where short-run MCMC samples have a realistic appearance and long-run MCMC samples have an unrealistic appearance. The work uses the term *convergent learning* to refer to the expected outcome where short-run and long-run MCMC samples have similar appearance, and the term *non-convergent learning* to refer to the unexpected but prevalent outcome where models have realistic short-run samples and oversaturated long-run samples. Convergent learning is essential for our defense strategy because long-run samples must be realistic so that classifiers can maintain high accuracy after Langevin transformation (see Section 4.1 and Appendix D).

As observed by Nijkamp et al. (2020), we were unable to learn a convergent model when updating $\theta$ with the Adam optimizer (Kingma & Ba, 2015). Despite the drawbacks of Adam for convergent learning, it is a very effective tool for obtaining realistic short-run synthesis. Drawing inspiration from classifier training from Keskar & Socher (2017), we learn a convergent EBM in two phases. The first phase uses Adam to update $\theta$ to achieve realistic short-run synthesis. The second phase uses SGD to update $\theta$ to align short-run and long-run MCMC samples to correct the degenerate steady-state from the Adam phase. This modification allows us to learn the convergent EBMs for complex datasets such as CIFAR-10 using 20% of the computational budget of Nijkamp et al. (2020). See Figure 2. We use the lightweight EBM from Nijkamp et al. (2019) as our network architecture.

We use long run chains for our EBM defense to remove adversarial signals while maintaining image features needed for accurate classification. The steady-state convergence property ensures adversarial signals will eventually vanish, while metastable behaviors preserve features of the initial state. Theoretical perspectives on our defense can be found in Appendix C and a comparison of convergent and non-convergent EBM defenses can be found in Section 4.1 and Appendix D.

# 3 ATTACK AND DEFENSE FORMULATION

## 3.1 CLASSIFICATION WITH STOCHASTIC TRANSFORMATIONS

Let $T(x)$ be a stochastic pre-processing transformation for a state $x \in \mathbb{R}^D$. Given a fixed input $x$, the transformed state $T(x)$ is a random variable over $\mathbb{R}^D$. In this work, $T(x)$ is obtained from $K$ steps of the Langevin update (3) starting from $X_0 = x$. One can compose $T(x)$ with a deterministic classifier $f(x) \in \mathbb{R}^J$ (for us, a naturally-trained classifier) to define a new classifier $F(x) \in \mathbb{R}^J$ as

$$F(x) = E_{T(x)}[f(T(x))]. \tag{4}$$

$F(x)$ is a deterministic classifier even though $T(x)$ is stochastic. The predicted label for $x$ is then $c(x) = \arg\max_j F(x)_j$. In this work, $f(x)$ denotes logits and $F(x)$ denotes expected logits, although other choices are possible (e.g. softmax outputs). We refer to (4) as an Expectation-Over-Transformation (EOT) defense. The classifier $F(x)$ in (4) is simply the target of the EOT attack (Athalye et al., 2018). The importance of the EOT formulation is well-established for adversarial attacks, but its importance for adversarial defense has not yet been established. Although direct evaluation of $F(x)$ is generally impossible, the law of large numbers ensures that the finite-sample approximation of $F(x)$ given by

$$\hat{F}_H(x) = \frac{1}{H} \sum_{h=1}^{H} f(\hat{x}_h) \quad \text{where,} \quad \hat{x}_h \sim T(x) \text{ i.i.d.,} \tag{5}$$

can approximate $F(x)$ to any degree of accuracy for a sufficiently large sample size $H$. In other words, $F(x)$ is intractable but trivial to accurately approximate via $\hat{F}_H(x)$ given enough computation.

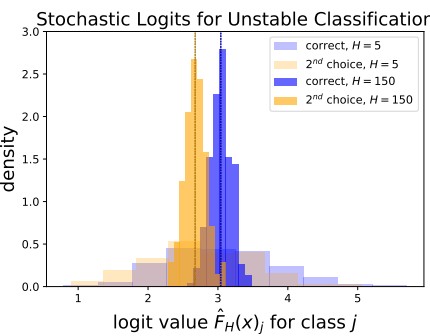 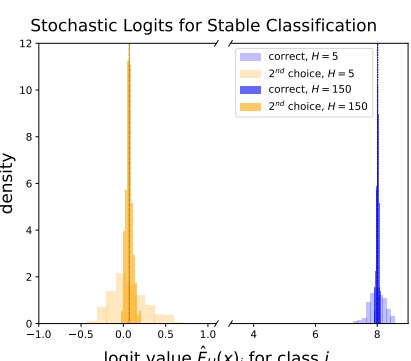

Figure 3: The histograms display different realizations of the logits $\hat{F}_H(x)$ for the correct class and the second most probable class for images $x_1$ (*left*) and $x_2$ (*right*) over different choices of $H$. In both cases, $F(x)$ (dashed vertical lines) gives correct classification. However, the overlap between the logit histograms of $\hat{F}_H(x_1)$ indicate a high probability of misclassification even for large $H$, while $\hat{F}_H(x_2)$ gives correct prediction even for small $H$ because the histograms are well-separated. The EOT defense formulation (4) is essential for securing borderline images such as $x_1$.

In the literature both attackers (Athalye et al., 2018; Tramer et al., 2020) and defenders (Dhillon et al., 2018; Yang et al., 2019; Grathwohl et al., 2020) evaluate stochastic classifiers of the form $f(T(x))$ using either $\hat{F}_1(x)$ or $\hat{F}_{H_{adv}}(x)$ where $H_{adv}$ is the number of EOT attack samples, typically around 10 to 30. This evaluation is not sound when $\hat{F}_H(x)$ has a small but plausible chance of misclassification because randomness alone could cause $x$ to be identified as an adversarial image even though $\hat{F}_H(x)$ gives the correct prediction on average (see Figure 3). In experiments with EBM defense, we identify many images $x$ that exhibit variation in the predicted label

$$\hat{c}_H(x) = \arg\max_j \hat{F}_H(x)_j \tag{6}$$

for smaller $H \approx 10$ but which have consistently correct prediction for larger $H \approx 150$. Fair evaluation of stochastic defenses must be based on the deterministic EOT defense $F$ in (4) and attackers must use sufficiently large $H$ to ensure that $\hat{F}_H(x)$ accurately approximates $F(x)$ before declaring that an attack using adversarial image $x$ is successful. In practice, we observe that $\hat{F}_{150}$ is sufficiently stable to evaluate $F$ over several hundred attacks with Algorithm 1 for our EBM defense.

## 3.2 ATTACKING STOCHASTIC CLASSIFIERS

Let $L(F(x), y) \in \mathbb{R}$ be the loss (e.g. cross-entropy) between a label $y \in \{1, \dots, J\}$ the outputs $F(x) \in \mathbb{R}^J$ of a classifier (e.g. the logits) for an image $x \in \mathbb{R}^D$. For a given pair of observed data $(x^+, y)$, an untargeted white-box adversarial attack searches for the state

$$x_{\mathrm{adv}}(x^+, y) = \arg\max_{x \in S} L(F(x), y) \tag{7}$$

that maximizes loss for predicting $y$ in a set $S \subset \mathbb{R}^D$ centered around $x^+$. In this work, a natural image $x^+$ will have pixels intensities from 0 to 1 (i.e. $x^+ \in [0,1]^D$). One choice of S is the intersection of the image hypercube $[0,1]^D$ and the $l_\infty$-norm $\varepsilon$-ball around $x^+$ for suitably small $\varepsilon > 0$. Another option is the intersection of the hypercube $[0,1]^D$ with the $l_2$-norm $\varepsilon$-ball.

The Projected Gradient Descent (PGD) attack (Madry et al., 2018) is the standard benchmark when $S$ is the $\varepsilon$-ball in the $l_p$ norm. PGD begins at a random $x_0 \in S$ and maximizes (7) by iteratively updating $x_i$ with

$$x_{i+1} = \prod_S (x_i + \alpha g(x_i, y)), \quad g(x, y) = \arg\max_{\|v\|_p \leq 1} v^\top \Delta(x, y), \tag{8}$$

where $\Pi_S$ denotes projection onto $S$, $\Delta(x, y)$ is the attack gradient, and $\alpha > 0$ is the attack step size. Standard PGD uses the gradient $\Delta_{\mathrm{PGD}}(x, y) = \nabla_x L(F(x), y)$.

The EOT attack (Athalye et al., 2018) circumvents the intractability of $F$ by attacking finite sample logits $\hat{F}_{H_{\mathrm{adv}}}$, where $H_{\mathrm{adv}}$ is the number of EOT attack samples, with the gradient

$$\Delta_{\mathrm{EOT}}(x, y) = \nabla_x L(\hat{F}_{H_{\mathrm{adv}}}(x), y). \tag{9}$$

The EOT attack is the natural adaptive attack for our EOT defense formulation. Another challenge when attacking a preprocessing defense is the computational infeasibility or theoretical impossibility of differentiating $T(x)$. The Backward Pass Differentiable Approximation (BPDA) technique (Athalye et al., 2018) uses an easily differentiable function $g(x)$ such that $g(x) \approx T(x)$ to attack $F(x) = f(T(x))$. One calculates the attack loss using $L(f(T(x)), y)$ on the forward pass but calculates the attack gradient using $\nabla_x L(f(g(x)), y)$ on the backward pass. A simple but effective form of BPDA is the identity approximation $g(x) = x$. This approximation is reasonable for preprocessing defenses that seek to remove adversarial signals while preserving the main features of the original image. When $g(x) = x$, the BPDA attack gradient is $\Delta_{\mathrm{BPDA}}(x, y) = \nabla_z L(f(z), y)$ where $z = T(x)$. Intuitively, this attack obtains an attack gradient with respect to the purified image and applies it to the original image.

Combining the EOT attack and BPDA attack with identity $g(x) = x$ gives the attack gradient

$$\Delta_{\mathrm{BPDA+EOT}}(x, y) = \frac{1}{H_{\mathrm{adv}}} \sum_{h=1}^{H_{\mathrm{adv}}} \nabla_{\hat{x}_h} L \left( \frac{1}{H_{\mathrm{adv}}} \sum_{h=1}^{H_{\mathrm{adv}}} f(\hat{x}_h), y \right), \quad \hat{x}_h \sim T(x) \text{ i.i.d.} \tag{10}$$

The BPDA+EOT attack represents the strongest known attack against preprocessing defenses, as shown by its effective use in recent works (Tramer et al., 2020). We use $\Delta_{\mathrm{BPDA+EOT}}(x, y)$ in (8) as our primary attack to evaluate the EOT defense (4). Pseudo-code for our adaptive attack can be found in Algorithm 1 in Appendix A.

## 4 EXPERIMENTS

We use two different network architectures in our experiments. The first network is the lightweight EBM from Nijkamp et al. (2019). The second network is a $28 \times 10$ Wide ResNet classifier (Zagoruyko

& Komodakis, 2016). The EBM and classifier are trained independently on the same dataset. No adversarial training or other training modifications are used for either model. The EBM defense in our work is given by Equation (4) and approximated by Equation (5), where $T(x)$ is $K$ steps of the Langevin iterations in Equation (3).We use the parameters from Algorithm 1 for all evaluations unless otherwise noted. In Section 4.1, we examine the effect of the number of Langevin steps $K$ and the stability of Langevin sampling paths on defense. Section 4.2 examines our defense against a PGD attack from the base classifier, and Section 4.3 examines our defense against the adaptive BPDA+EOT attack.

## 4.1 COMPARISON OF CONVERGENT AND NON-CONVERGENT DEFENSES

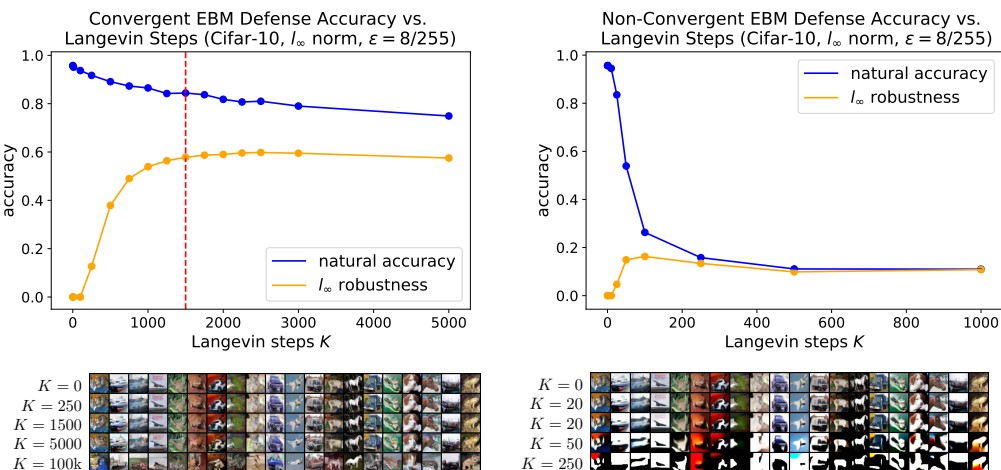

Figure 4: Accuracy on natural images and adversarial images from a BPDA+EOT attack (10) for EBM defense with different number of Langevin steps, and images sampled from the EBM. *Left:* Defense with a convergent EBM. Using approximately 1500 Langevin steps yields a good balance of natural and robust accuracy. *Right:* Defense with non-convergent EBM. Oversaturated long-run images prevent non-convergent EBM defense from achieving high natural or robust accuracy.

We examine the effect the number of Langevin steps has on defense accuracy for the CIFAR-10 dataset (see Figure 4). Each point displays either the baseline accuracy of our stochastic classifier (blue) or the results of a BPDA+EOT attack (orange) on 1000 test images. The attacks used to make this diagram use a reduced load of $H_{\text{adv}} = 7$ replicates for EOT attacks so these defense accuracies are slightly higher than the full attack results presented in Figure 5. Short-run Langevin with $K \leq 100$ steps yields almost no adversarial robustness. Increasing the number of steps gradually increases robustness until the defense saturates at around $K = 2000$. We chose $K = 1500$ steps in our experiments as a good tradeoff between robustness, natural accuracy, and computational cost.

For comparison, we run the same experiment using a non-convergent EBM. The network structure and training are identical to our convergent model, except that we use Adam instead of SGD throughout training. The non-convergent EBM defense cannot achieve high natural accuracy with long-run sampling because of the oversaturated features that emerge. Without a high natural accuracy, it is impossible to obtain good defense results. Thus convergent EBMs that can produce realistic long-run samples are a key ingredient for the success of our method. In the context of EBM defense, short-run sampling refers to trajectories of up to about 100 steps where no defensive properties emerge, and long-run sampling refers to trajectories of about 1000 or more steps where defense becomes possible.

## 4.2 PGD ATTACK FROM BASE CLASSIFIER FOR CIFAR-10 DATASET

We first evaluate our defense using adversarial images created from a PGD attack on the classifier $f(x)$. Since this attack does not incorporate the Langevin sampling from $T(x)$, the adversarial images in this section are relatively easy to secure with Langevin transformations. This attack serves as a benchmark for comparing our defense to the IGEBM (Du & Mordatch, 2019) and JEM (Grathwohl

et al., 2020) models that also evaluate adversarial defense with a ConvNet EBM (1). For all methods, we evaluate the base classifier and the EBM defense for $K = 10$ Langevin steps (as in prior defenses) and $K = 1500$ steps (as in our defense). The results are displayed in Table 1.

Table 1: CIFAR-10 accuracy for our EBM defense and prior EBM defenses against a PGD attack from the base classifier $f(x)$ with $l_\infty$ perturbation $\varepsilon = 8/255$. (*evaluated on 1000 images*)

| | Base Classifier $f(x)$ | | EBM Defense, $K = 10$ | | EBM Defense, $K = 1500$ | |
|---|---|---|---|---|---|---|
| | Nat. | Adv. | Nat. | Adv. | Nat. | Adv. |
| **Ours** | 0.9530 | 0.0000 | 0.9586 | 0.0001 | 0.8412 | 0.7891 |
| (Du & Mordatch, 2019) | 0.4714 | 0.3219 | 0.4885 | 0.3674 | 0.487* | 0.375* |
| (Grathwohl et al., 2020) | 0.9282 | 0.0929 | 0.9093 | 0.1255 | 0.755* | 0.238* |

Our natural classifier $f(x)$ has a high base accuracy but no robustness. The JEM base classifier has high natural accuracy and minor robustness, while the IGEBM base classifier has significant robustness but very low natural accuracy. Short-run sampling with $K = 10$ Langevin steps does not significantly increase robustness for any model. Long-run sampling with $K = 1500$ steps provides a dramatic increase in defense for our method but only minor increases for the prior methods. Further discussion of the IGEBM and JEM defenses can be found in Appendix G.

## 4.3 BPDA+EOT ATTACK

In this section, we evaluate our EBM defense using the adaptive BPDA+EOT attack (10) designed specifically for our defense approach. This attack is recently used by Tramer et al. (2020) to evaluate the preprocessing defense from Yang et al. (2019) that is similar to our method.

Table 2: Defense vs. whitebox attacks with $l_\infty$ perturbation $\varepsilon = 8/255$ for CIFAR-10.

| Defense | $f(x)$ Train Ims. | $T(x)$ Method | Attack | Nat. | Adv. |
|---|---|---|---|---|---|
| **Ours** | Natural | Langevin | BPDA+EOT | 0.8412 | 0.5490 |
| (Madry et al., 2018) | Adversarial | – | PGD | 0.873 | 0.458 |
| (Zhang et al., 2019) | Adversarial | – | PGD | 0.849 | 0.5643 |
| (Carmon et al., 2019) | Adversarial | – | PGD | 0.897 | 0.625 |
| (Song et al., 2018) | Natural | Gibbs Update | BPDA | 0.95 | 0.09 |
| (Srinivasan et al., 2019) | Natural | Langevin | PGD | – | 0.0048 |
| (Yang et al., 2019) | Transformed | Mask + Recon. | BPDA+EOT | 0.94 | 0.15 |

**CIFAR-10**. We ran 5 random restarts of the BPDA+EOT attack in Algorithm 1 with the the listed parameters on the entire CIFAR-10 test set. In particular, the attacks use adversarial perturbation $\varepsilon = 8/255$ and attack step size $\alpha = 2/255$ in the $l_\infty$ norm. One evaluation of the entire test set took approximately 2.5 days using 4x RTX 2070 Super GPUs. We compare our results to a representative selection of AT and AP defenses in Table 2. We include the training method for the classifier, preprocessing transformation (if any), and the strongest attack for each defense.

Our EBM defense is stronger than standard AT (Madry et al., 2018) and comparable to modified AT from Zhang et al. (2019). Although our results are not on par with state-of-the-art AT (Carmon et al., 2019), our defense is the first method that can effectively secure naturally-trained classifiers.

We now examine the effect of the perturbation $\varepsilon$, number of attacks $N$, and number of EOT attack replicates $H_{\text{adv}}$ on the strength of the BPDA+EOT attack. To reduce the computational cost of the diagnostics, we use a fixed set of 1000 randomly selected test images for diagnostic attacks.

Figure 5 displays the robustness of our model compared against two AT models, standard AT (Madry et al., 2018) and Semi-supervised AT (Carmon et al., 2019) for $l_\infty$ and $l_2$ attacks across perturbation size $\varepsilon$. Our model is attacked with BPDA+EOT while AT models are attacked with PGD. Our defense is more robust than standard AT for a range of medium-size perturbations in the $l_\infty$ norm and much

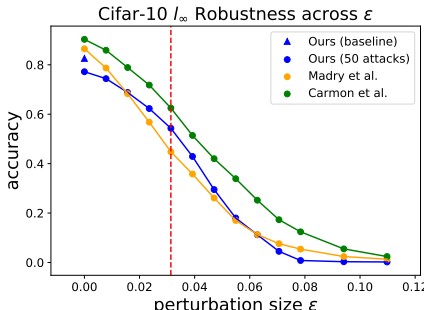
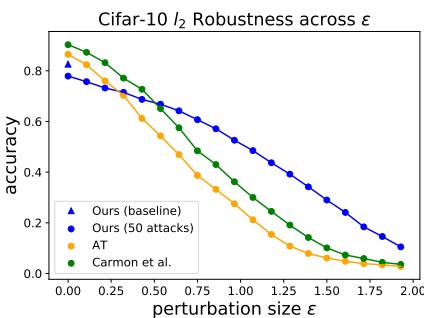

Figure 5: Accuracy across perturbation $\varepsilon$ for $l_\infty$ and $l_2$ attacks against our defense, standard AT (Madry et al., 2018) and Semi-supervised AT (Carmon et al., 2019).

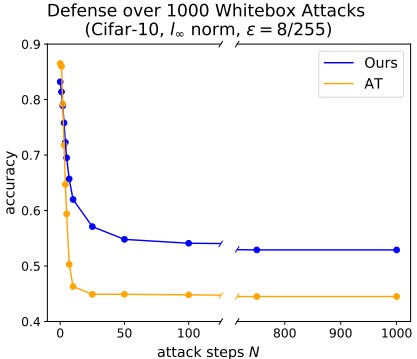
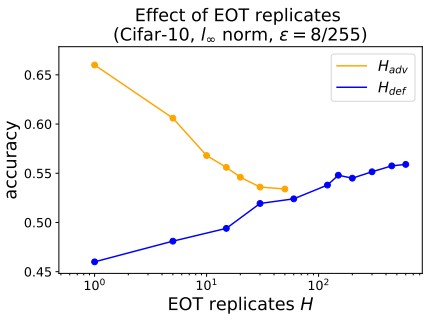

Figure 6: Effect of number of attack steps $N$ and number of EOT replicates $H_{\text{adv}}$ and $H_{\text{def}}$.

more robust than standard AT for medium and large $\varepsilon$ in the $l_2$ norm. Our model is less robust than Semi-supervised AT for the $l_\infty$ norm but more robust for the $l_2$ norm. AT-based models are trained to defend against a specific attack while our method does not use any attacks during training. This is likely a reason why our defense outperforms $l_\infty$-trained AT against $l_2$ attacks.

Figure 6 visualizes the effect of increasing the computational power of the attacker and defender in Algorithm 1. The left figure compares our defense with AT over 1000 attacks using an increased number $H_{\text{adv}} = 20$ of attack replicates. The majority of breaks happen within the first 50 attacks as used in the CIFAR-10 experiment in Table 2, while a small number of breaks occur within a few hundred attack steps. It is likely that some breaks from long-run attacks are the result of lingering stochastic behavior from $\hat{F}_{H_{\text{def}}}(x)$ rather than the attack itself. The right figure shows the effect of the number of EOT replicates. The strength of the EOT attack saturates when using 20 to 30 replicates. A small gap in attack strength remains between the 15 replicates used in our attacks and the strongest possible attack. Some of this effect is likely mitigated by our use of 5 random restarts. Defense with $H_{\text{def}} = 150$ is close to the maximum achievable defense for our method, although more replicates would slightly strengthen our results.

The evaluation in Table 2 pushes the limits of what is computationally feasible with widely available resources. The diagnostics indicate that our defense report is an accurate approximation of the defense of our ideal classifier $F(x)$ in (4) against the BPDA+EOT attack (10), although more computation would yield a slightly more accurate estimate.

**SVHN and CIFAR-100**. The attack and defense parameters for our method are identical to those used in the CIFAR-10 experiments. We compare our results with standard AT. Overall, our defense performs well for datasets that are both simpler and more complex than CIFAR-10. In future work, further stabilization of image appearance after Langevin sampling could yield significant benefits for settings where precise details need to be preserved for accurate classification. The AT results for CIFAR-100 are from Balaji et al. (2019) and the results for SVHN are from our implementation.

Table 3: Defense vs. whitebox attacks with $l_\infty$ perturbation $\varepsilon = 8/255$ for SVHN and CIFAR-100.

| | SVHN | | CIFAR-100 | |
|---|---|---|---|---|
| | Nat. | Adv. | Nat. | Adv. |
| **Ours** | 0.9223 | 0.6755 | 0.5166 | 0.2610 |
| (Madry et al., 2018) | 0.8957 | 0.5039 | 0.5958 | 0.2547 |

## 5 RELATED WORK

**Adversarial training** learns a robust classifier using adversarial images created during each weight update. The method is introduced by Madry et al. (2018). Many variations of AT have been explored, some of which are related to our defense. He et al. (2019) apply noise injection to each network layer to increase robustness via stochastic effects. Similarly, Langevin updates with our EBM can be interpreted as a ResNet (He et al., 2016) with noise injected layers as discussed by Nijkamp et al. (2019). Semi-supervised AT methods (Alayrac et al., 2019; Carmon et al., 2019) use unlabeled images to improve robustness. Our EBM also leverages unlabeled data for defense.

**Adversarial preprocessing** is a strategy where auxiliary transformations are applied to adversarial inputs before they are given to the classifier. Prior forms of pre-processing defenses include rescaling (Xie et al., 2018a), thermometer encoding (Buckman et al., 2018), feature squeezing (Xu et al., 2018), activation pruning (Dhillon et al., 2018), reconstruction (Samangouei et al., 2018), ensemble transformations (Raff et al., 2019), addition of Gaussian noise (Cohen et al., 2019), and reconstruction of masked images (Yang et al., 2019). Prior preprocessing methods also include energy-based models such as Pixel-Defend (Song et al., 2018) and MALADE (Srinivasan et al., 2019) that draw samples from a density that differs from (1). It was shown by Athalye et al. (2018); Tramer et al. (2020) that many preprocessing defenses can be totally broken or dramatically weakened by simple adjustments to the standard PGD attack, namely the EOT and BPDA techniques. Furthermore, many preprocessing defenses such as Cohen et al. (2019); Raff et al. (2019); Yang et al. (2019) also modify classifier training so that the resulting defenses are analogous to AT, as discussed in Appendix F. A concurrent work called Denoised Smoothing (Salman et al., 2020) prepends a Denoising Autoencoder to a natural image classifier. The joint classifier fixes the natural classifier weights and the denoiser is trained by minimizing the cross-entropy loss of the joint model using noisy training inputs. Our defensive transformation provides security without information from any classifier. No prior preprocessing defense is competitive with AT when applied to independent naturally-trained classifiers.

**Energy-based models** are a probabilistic method for unsupervised modeling. Early energy-based image models include the FRAME (Zhu et al., 1998) and RBM (Hinton, 2002). The EBM is a descriptive model (Guo et al., 2003; Zhu, 2003). Our EBM (1) is introduced as the DeepFRAME model by Xie et al. (2016) and important observations about the learning process are presented by Nijkamp et al. (2020; 2019). Other studies of EBM learning include Gao et al. (2018); Xie et al. (2018b); Gao et al. (2020). Preliminary investigations for using the EBM (1) for adversarial defense are presented by Du & Mordatch (2019); Grathwohl et al. (2020) but the results are not competitive with AT (see Section 4.2). Our experiments show that the instability of sampling paths for these non-convergent models prevents Langevin trajectories from manifesting defensive properties. We build on the convergent learning methodology from Nijkamp et al. (2020) to apply long-run Langevin sampling as a defense technique.

## 6 CONCLUSION

This work demonstrates that Langevin sampling with a convergent EBM is an effective defense for naturally-trained classifiers. Our defense is founded on an improvement to EBM training that enables efficient learning of stable long-run sampling for complex datasets. We evaluate our defense using non-adaptive and adaptive whitebox attacks for the CIFAR-10, CIFAR-100, and SVHN datasets. Our defense is competitive with adversarial training. Securing naturally-trained classifiers with post-training defense is a long-standing open problem that this work resolves.

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

# A   ATTACK ALGORITHM

---

**Algorithm 1** BPDA+EOT adaptive attack to evaluate EOT defense (4)

---

**Require:** Natural images $\{x_m^+\}_{m=1}^M$, EBM $U(x)$, classifier $f(x)$, Langevin noise $\tau = 0.01$, Langevin updates $K = 1500$, number of attacks $N = 50$, attack step size $\alpha = \frac{2}{255}$, maximum perturbation size $\varepsilon = \frac{8}{255}$, EOT attack samples $H_{\text{adv}} = 15$, EOT defense samples $H_{\text{def}} = 150$

**Ensure:** Defense record $\{d_m\}_{m=1}^M$ for each image.

  **for** m=1:M **do**
    Calculate large-sample predicted label of the natural image $\hat{c}_{H_{\text{def}}}(x_m^+)$ with (6).
    **if** $\hat{c}_{H_{\text{def}}}(x_m^+) \neq y_m$ **then**
      Natural image misclassified. $d_m \leftarrow$ `False`. End loop iteration $m$.
    **else**
      $d_m \leftarrow$ `True`.
    **end if**
    Randomly initialize $X_0$ in the $l_p$ $\varepsilon$-ball centered at $x_m^+$ and project to $[0,1]^D$.
    **for** n=1:(N+1) **do**
      Calculate small-sample predicted label $\hat{c}_{H_{\text{adv}}}(X_{n-1})$ with (6).
      Calculated attack gradient $\Delta_{\text{BPDA+EOT}}(X_{n-1}, y_m)$ with (10).
      **if** $\hat{c}_{H_{\text{adv}}}(X_{n-1}) \neq y_m$ **then**
        Calculate large-sample predicted label $\hat{c}_{H_{\text{def}}}(X_{n-1})$ with (6).
        **if** $\hat{c}_{H_{\text{def}}}(X_{n-1}) \neq y_m$ **then**
          The attack has succeeded. $d_m \leftarrow$ `False`. End loop iteration $m$.
        **end if**
      **end if**
      Use $\Delta_{\text{BPDA+EOT}}(X_{n-1}, y_m)$ with the $l_p$ $\varepsilon$-bounded PGD update (8) to obtain $X_n$.
    **end for**
  **end for**

---

Algorithm 1 is pseudo-code for the attack and defense framework described in Section 3. One notable aspect of the algorithm is the inclusion of an EOT defense phase to verify potentially successful attacks. Images which are identified as broken for the smaller sample used to generate EOT attack gradients are checked again using a much larger EOT defense sample to ensure that the break is due to the adversarial state and not random finite-sample effects. This division is done for purely computational reasons. It is extremely expensive to use 150 EOT attack replicates but much less expensive to use 15 EOT attack replicates as a screening method and to carefully check candidates for breaks when they are identified from time to time using 150 EOT defense replicates. In our experiments we find that the EOT attack is close to its maximum strength after about 15 to 20 replicates are used, while about 150 EOT defense replicates are needed for consistent evaluation of $F$ from (4) over several hundred attacks. Ideally, the same number of chains should be used for both EOT attack and defense, in which case the separate verification phase would not be necessary.

# B   IMPROVED LEARNING OF CONVERGENT EBMS

Algorithm 2 provides pseudo-code for our improvement of the convergent learning method from Nijkamp et al. (2020). This implementation allows efficient learning of convergent EBMs for complex datasets.

# C   ERASING ADVERSARIAL SIGNALS WITH MCMC SAMPLING

This section discusses two theoretical perspectives that justify the use of an EBM for purifying adversarial signals: memoryless and chaotic behaviors from sampling dynamics. We emphasize that the discussion applies primarily to long-run behavior of a Langevin image trajectory. Memoryless and chaotic properties do not appear to emerge from short-run sampling. Throughout our experiments, we never observe significant defense benefits from short-run Langevin sampling.

---

**Algorithm 2** ML with Adam to SGD Switch for Convergent Learning of EBM (1)

---

**Require:** ConvNet potential $U(x; \theta)$, number of training steps $J = 150000$, step to switch from SGD to Adam $J_{\text{SGD}} = 50000$, initial weight $\theta_1$, training images $\{x_i^+\}_{i=1}^{N_{\text{data}}}$, data perturbation $\tau_{\text{data}} = 0.02$, step size $\tau = 0.01$, Langevin steps $K = 100$, Adam learning rate $\gamma_{\text{Adam}} = 0.0001$, SGD learning rate $\gamma_{\text{SGD}} = 0.00005$.
**Ensure:** Weights $\theta_{J+1}$ for energy $U(x; \theta)$.

Set optimizer $g \leftarrow \text{Adam}(\gamma_{\text{Adam}})$. Initialize persistent image bank as $N_{\text{data}}$ uniform noise images.
**for** $j$=1:($J$+1) **do**
  **if** $j = J_{\text{SGD}}$ **then**
    Set optimizer $g \leftarrow \text{SGD}(\gamma_{\text{SGD}})$.
  **end if**
  1. Draw batch images $\{x_{(i)}^+\}_{i=1}^m$ from training set, where $(i)$ indicates a randomly selected index for sample $i$, and get samples $X_i^+ = x_{(i)} + \tau_{\text{data}}Z_i$, where $Z_i \sim \text{N}(0, I_D)$ i.i.d.
  2. Draw initial negative samples $\{Y_i^{(0)}\}_{i=1}^m$ from persistent image bank. Update $\{Y_i^{(0)}\}_{i=1}^m$ with the Langevin equation

$$Y_i^{(k)} = Y_i^{(k-1)} - \frac{\tau^2}{2}\frac{\partial}{\partial y}U(Y_i^{(k-1)}; \theta_j) + \tau Z_{i,k},$$

  where $Z_{i,k} \sim \text{N}(0, I_D)$ i.i.d., for $K$ steps to obtain samples $\{X_i^-\}_{i=1}^m = \{Y_i^{(K)}\}_{i=1}^m$. Update persistent image bank with images $\{Y_i^{(K)}\}_{i=1}^m$.
  3. Update the weights by $\theta_{j+1} = \theta_j - g(\Delta\theta_j)$, where $g$ is the optimizer and

$$\Delta\theta_j = \frac{\partial}{\partial\theta}\left(\frac{1}{n}\sum_{i=1}^n U(X_i^+; \theta_j) - \frac{1}{m}\sum_{i=1}^m U(X_i^-; \theta_j)\right)$$

  is the ML gradient approximation.
**end for**

---

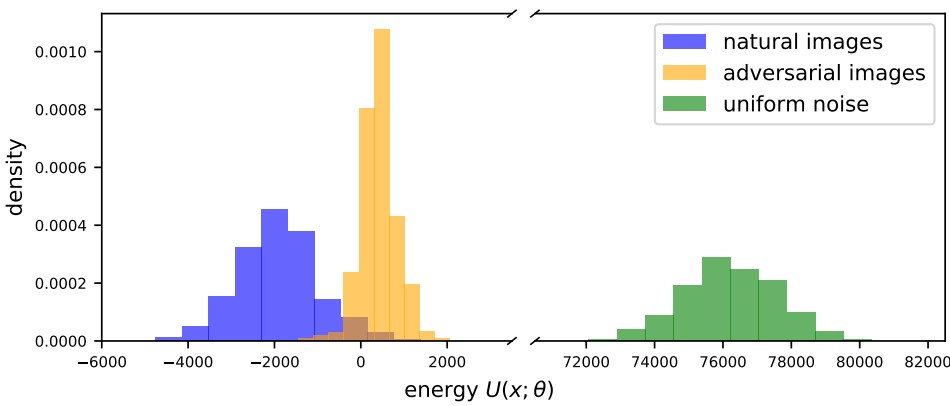

Figure 7: Energy $U(x; \theta)$ of natural, adversarial, and noise images.

## C.1 MEMORYLESS DYNAMICS

The first justification of EBM defense is that iterative probabilistic updates will move an image from a low-probability adversarial region to a high-probability natural region, as discussed in prior works (Song et al., 2018; Srinivasan et al., 2019; Grathwohl et al., 2020). Comparing the energy of adversarial and natural images shows that adversarial images tend to have marginally higher energy,

which is evidence that adversarial images are improbable deviations from the steady-state manifold of the EBM that models the distribution of natural images (see Figure 7).

The theoretical foundation for removing adversarial signals with MCMC sampling comes from the well-known steady-state convergence property of Markov chains. The Langevin update (3) is designed to converge to the distribution $p(x; \theta)$ learned from unlabeled data after an infinite number of Langevin steps. The steady-state property guarantees that adversarial signals will be erased from the sampled state as long as enough steps are used because the sampled state will have no dependence on the initial state. This property is actually too extreme because full MCMC mixing would completely undermine classification by causing samples to jump between class modes.

The quasi-equilibrium and metastable behaviors of MCMC sampling (Bovier & den Hollander, 2006) can be as useful as its equilibrium properties. Metastable behavior refers to intramodal mixing that can occur for MCMC trajectories over intermediate time-scales, in contrast to the limiting behavior of full intermodal mixing that occurs for trajectories over arbitrarily large time-scales. Although slow-mixing and high autocorrelation of MCMC chains are often viewed as major shortcomings, these properties enable EBM defense by preserving class-specific features while sampling erases adversarial signals.

Our EBM samples always exhibit some dependence on the initial state for computationally feasible Langevin trajectories. Mixing within a metastable region can greatly reduce the influence of an initial adversarial signal even when full mixing is not occurring. Successful classification of long-run MCMC samples occurs when the metastable regions of the EBM $p(x; \theta)$ are aligned with the class boundaries learned by the classifier network $f(x)$. Our experiments show that this alignment naturally occurs for convergent EBMs and naturally-trained classifiers. No training modification for $f(x)$ is needed to correctly classify long-run EBM samples. Our defense relies on a balance between the memoryless properties of MCMC sampling that erase noise and the metastable properties of MCMC sampling that preserve the initial state.

## C.2 Chaotic Dynamics

Chaos theory provides another perspective for justifying the erasure of adversarial signals with long-run iterative transformations. Intuitively, a deterministic system is chaotic if an initial infinitesimal perturbation grows exponentially in time so that paths of nearby points become distant as the system evolves (i.e. the butterfly effect). The same concept can be extended to stochastic systems. The SDE

$$\frac{dX}{dt} = V(X) + \eta\xi(t), \tag{11}$$

where $\xi(t)$ is Brownian motion and $\eta \geq 0$, that encompasses the Langevin equation is known to exhibit chaotic behavior in many contexts for sufficiently large $\eta$ (Lai et al., 2003).

One can determine whether a dynamical system is chaotic or ordered by measuring the maximal Lyapunov exponent $\lambda$ given by

$$\lambda = \lim_{t\to\infty} \frac{1}{t} \log \frac{|\delta X_\eta(t)|}{|\delta X_\eta(0)|} \tag{12}$$

where $\delta X_\eta(t)$ is an infinitesimal perturbation between system state at time $t$ after evolution according to (11) from an initial perturbation $\delta X_\eta(0)$. For ergodic dynamics, $\lambda$ does not depend on the initial perturbation $\delta X_\eta(0)$. Ordered systems have a maximal Lyapunov exponent that is either negative or 0, while chaotic systems have positive Lyapunov exponents. The SDE (11) will have a maximal exponent of at least 0 since dynamics in the direction of gradient flow are neither expanding nor contracting. One can therefore detect whether a Langevin equation yields ordered or chaotic dynamics by examining whether its corresponding maximal Lyapunov exponent is 0 or positive.

We use the classical method of Benettin et al. (1976) to calculate the maximal Lyapunov exponent of the altered form Langevin transformation (3) given by

$$T_\eta(X) = X - \frac{\tau^2}{2} \nabla_X U(X; \theta) + \eta\tau Z_k \tag{13}$$

for a variety of noise strengths $\eta$. Our results exhibit the predicted transition from noise to chaos. The value $\eta = 1$ which corresponds to our training and defense algorithms is just beyond the transition

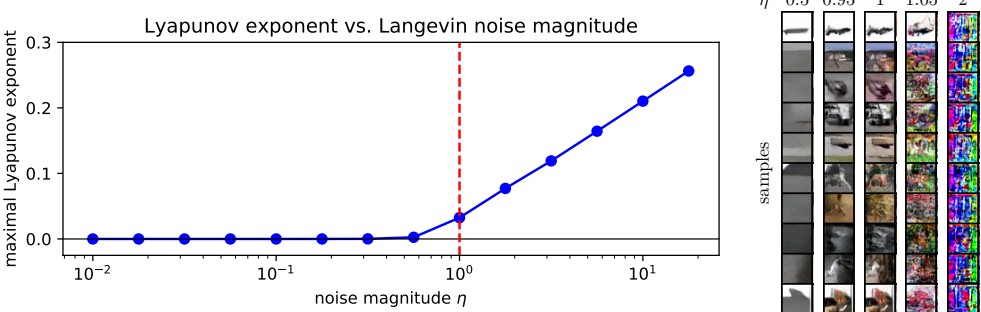

Figure 8: *Left:* Maximal Lyapunov exponent for different values of $\eta$. The value $\eta = 1$ which corresponds to our training and defense sampling dynamics is just above the transition from the ordered region where the maximal exponent is 0 to the chaotic region that where the maximal exponent is positive. *Right:* Appearance of steady-state samples for different values of $\eta$. Oversaturated images appear for low values of $\eta$, while noisy images appear for high $\eta$. Realistic synthesis is achieved in a small window around $\eta = 1$ where gradient and noise forces are evenly balanced.

from the ordered region to the chaotic region. Our defense dynamics occur in a critical interval where ordered gradient forces that promote pattern formation and chaotic noise forces that disrupt pattern formation are balanced. Oversaturation occurs when the gradient dominates and noisy images occur when the noise dominates. The results are shown in Figure 8.

The unpredictability of paths under $T_\eta$ is an effective defense against BPDA because informative attack gradients cannot be generated through chaotic transformation. Changes in adversarial perturbation from one BPDA attack to the next attack are magnified by the Langevin transformation and it becomes difficult to climb the loss landscape $L(F(x), y)$ to create adversarial samples. Other chaotic transformations, either stochastic or deterministic, might be an interesting line of research as a class of defense methods.

## D  EFFECT OF NUMBER OF LANGEVIN STEPS ON FID SCORE

We examine the difference between convergent and non-convergent EBM sampling paths by measuring the FID score (Heusel et al., 2017) of samples initialized from the data (see Figure 9). The convergent EBM maintains a reasonably low FID score across many steps so that long-run samples can be accurately classified. The non-convergent EBM experiences a quick increase in FID as oversaturated samples appear. Labels for the oversaturated samples cannot be accurately predicted by a naturally-trained classifier, preventing successful defense.

## E  DISCUSSION OF DEFENSE RUNTIME

Robustness does not depend on the computational resources of the attacker or defender (Athalye et al., 2018). Nonetheless, we took efforts to reduce the computational cost of our defense to make it as practical as possible. Our defense requires less computation for classifier training but more computation for evaluation compared to AT due to the use of 1500 Langevin steps as a preprocessing procedure. Running 1500 Langevin steps on a batch of 100 images with our lightweight EBM takes about 13 seconds on a RTX 2070 Super GPU. While this is somewhat costly, it is still possible to evaluate our model on large test sets in a reasonable amount of time. The cost of our defense also poses a computational obstacle to attackers, since the BPDA+EOT attack involves iterative application of the defense dynamics. Thoroughly evaluating our defense on CIFAR-10 took about 2.5 days for the entire testing set with 4 RTX 2070 Super GPUs.

Our EBM is significantly smaller and faster than previous EBMs used for adversarial defense. Our EBM has less than 700K parameters and sampling is about $20\times$ to $30\times$ faster than IGEBM and JEM, as shown in Appendix G. Generating adversarial images using PGD against a large base classifier is also expensive (about $30\times$ slower for a PGD step compared to a Langevin step because our base

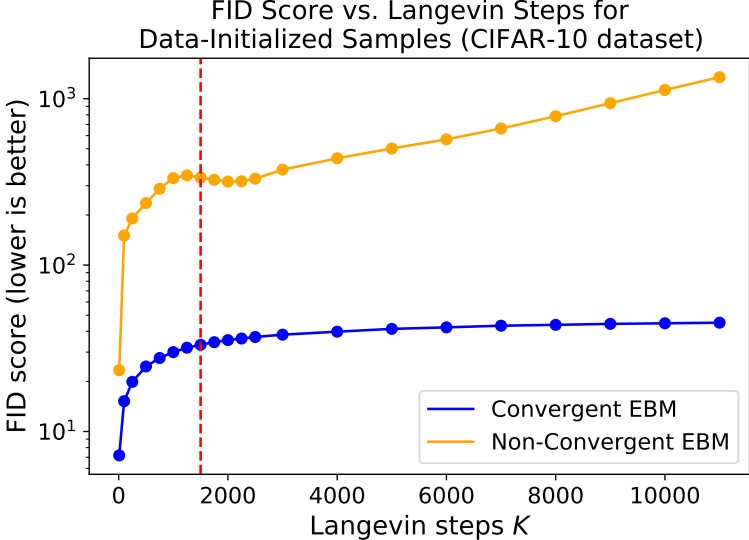

Figure 9: FID scores of samples from a convergent and non-convergent EBM that are initialized from training data. The FID of the convergent model remains reasonably low because the steady-state of the convergent model is aligned with the data distribution. The FID of the non-convergent model quickly increases because the oversaturated steady-state of the non-convergent model differs significantly from the data distribution. Maintaining a sampling distribution close to the data distribution is essential for achieving high prediction accuracy of a natural classifier on transformed states.

classifier uses the same Wide ResNet backbone as the JEM model). Therefore 50 PGD steps against the base classifier takes about the same time as 1500 Langevin steps with our EBM, so our model can defend images at approximately the same rate as an attacker who generates adversarial samples against the base classifier. Generating adversarial examples using BPDA+EOT is much slower because defense dynamics must be incorporated. Further reducing the cost of our defense is an important direction for future work.

Our approach is amenable toward massive parallelization. One strategy is to distribute GPU batches temporally and run our algorithm in high FPS with a time delay. Another strategy is to distribute the workload of computing $\hat{F}_H$ across parallel GPU batches.

## F  A NOTE ON MODIFIED CLASSIFIER TRAINING FOR PREPROCESSING DEFENSES

Many works that report significant robustness via defensive transformations (Cohen et al. (2019); Raff et al. (2019); Yang et al. (2019) and others) also modify classifier learning by training with transformed images rather than natural images. Prior defensive transformations that are strong enough to remove adversarial signals have the side effect of greatly reducing the accuracy of a naturally-trained classifier. Therefore, signals introduced by these defensive transformations (high Gaussian noise, ensemble transformations, ME reconstruction, etc.) can be also considered "adversarial" signals (albeit perceptible ones) because they heavily degrade natural classifier accuracy much like the attack signals. From this perspective, modifying training to classify transformed images is a direct analog of AT where the classifier is trained to be robust to "adversarial" signals from the defensive transformation itself rather than a PGD attack. Across many adversarial defense works, identifying a defensive transformation that removes attack signals while preserving high accuracy of a natural classifier has universally proven elusive.

Our experiments train the classifier with natural images alone and not with images generated from our defensive transformation (Langevin sampling). Our approach represents the first successful defense

based purely on transformation and validates an entirely different approach compared to defenses which modify training of the base classifier (Madry et al. (2018); Cohen et al. (2019); Raff et al. (2019); Yang et al. (2019) etc.). To our knowledge, showing that natural classifiers can be secured with post-training defensive transformation is a contribution that is unique in the literature. Our task independent approach has the potential of securing images for many applications using a single defense model, while AT and relatives must learn a robust model for each application.

# G    DISCUSSION OF IGEBM AND JEM DEFENSES

We hypothesize that the non-convergent behavior of the IGEBM (Du & Mordatch, 2019) and JEM (Grathwohl et al., 2020) models limits their use as an EBM defense method. Long-run samples from both models have oversaturated and unrealistic appearance (see Figure 10). Non-convergent learning behavior is a consequence of training implementation rather than model formulation. Convergent learning may be a path to robustness using the IGEBM and JEM defense methods.

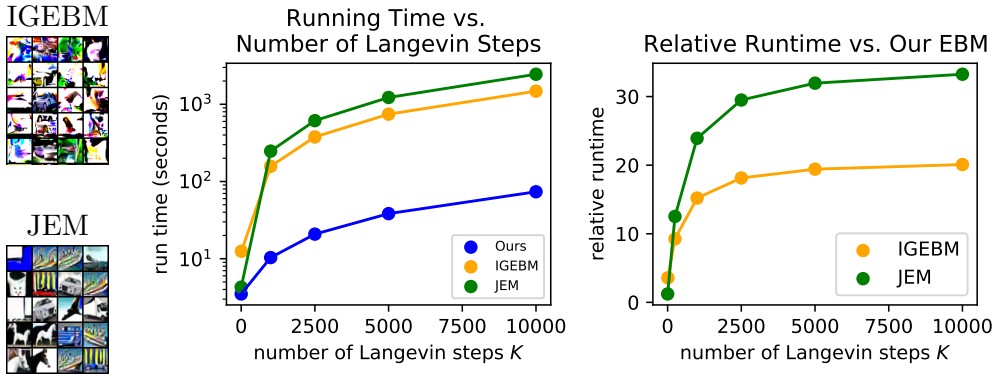

Figure 10: *Left:* Approximate steady-state samples of the IGEM and JEM models. Both exhibit oversaturation from non-convergent learning that can interfere with defense capabilities. *Right:* Comparison of running time for Langevin sampling with a batch of 100 images. The small scale and fast sampling of our EBM are important for the computational feasibility of our defense.

Both prior works use very large networks to maximize scores on generative modeling metrics. As a result, sampling from these models can be up to 30 times slower than sampling from our lightweight EBM structure from Nijkamp et al. (2019) (see Figure 10). The computational feasibility of our method currently relies on the the small scale of our EBM. Given the effectiveness of the weaker and less expensive PGD attack in Section 4.2 and the extreme computational cost of sampling with large EBM models, we do not to apply BPDA+EOT to the IGEBM or JEM defense.

The original evaluations of the IGEBM and JEM model use end-to-end backpropagation through the Langevin dynamics when generating adversarial examples. On the other hand, the relatively weak attack in Section 4.2 is as strong or much stronger than the theoretically ideal end-to-end attack. Gradient obfuscation from complex second-order differentiation might hinder the strength of end-to-end PGD when attacking Langevin defenses.

The IGEBM defense overcomes oversaturation by restricting sampling to a ball around the input image, but this likely prevents sampling from being able to manifest its defensive properties. An adversarial signal will be partially preserved by the boundaries of the ball regardless of how many sampling steps are used. Unrestricted sampling, as performed in our work and the JEM defense, is essential for removing adversarial signals.

