# OpenReview forum: "Stochastic Security: Adversarial Defense Using Long-Run Dynamics of Energy-Based Models"
_ICLR.cc/2021/Conference — ICLR 2021 Poster_

### Official Review · AnonReviewer2 · 2020-10-27
**The proposal in this paper is interesting. However, I have some concerns whether the proposals are as innovative as claimed in the paper.**

**Rating:** 7
**Confidence:** 5

**Review:**

This paper proposes a Langevin dynamics based approach to defend naturally trained classifiers. To perform Langevin dynamics, an energy based model (EBM) is utilized, which is trained with realistic long run MCMC samples for effective purification and hence defense.

The proposal in this paper are two fold:
1. Training of the EBM with long run dynamics along with using Adam optimizer in the initial stage and SGD for further training.
2. An Expectation over Transformation (EOT) defense based on Langevin dynamics.

+ves:
- The convergent learning of EBM was observed by Nijkamp et al. (2020) showing that the under this scenario short run and long-run MCMC samples have a similar appearance.
Similarly, Keskar and Soccer (2017) show that switching to SGD from Adam can provide improved convergence.
This paper utilizes the above mentioned ideas for training convergent EBM by using long run dynamics and by switching from Adam to SGD.

Although, the proposal in this paper is interesting, I have some concerns whether the proposals are as innovative as claimed in the paper.

Concerns:
- The main innovative part of the paper is that it used long run Langevin dynamics using EBM for defending naturally trained classifiers.
However Srinivasan et al. (2019) also utilized Langevin dynamics using EBM for the same task of defending against adversarial attacks, although the dynamics was performed with a modified distribution. A BPDA+EOT attack was also formulated to attack their own method in the above mentioned paper.
The authors do not make a clear distinction of how their method is different from the above mentioned paper.

- Although Adam was found not useful and switching from Adam to SGD was found to be helpful, the images from SGD alone is not shown.

- There is no clear definition of what is considered as a short-run and what is considered as a long-run dynamics.

- The Figure 3 lacks a detailed explanation. Are original images used in both the left and right figures or are they adversarial samples. Why is H=5,100 considered and not 150 as used in the experiments?

- Experimental settings in Table 1: JEM and IGEBM are also checked with long run dynamics. However, are the reported performance obtained by training them with long run dynamics? I suppose JEM and IGEBM can also suffer from unrealistic samples at long run due to non convergent learning as they are also EBM. In this case, the fair comparison should have been made by training both models with long run dynamics and switching from Adam to SGD.

- The most interesting part of the paper, in my humble opinion, is the effect of long run dynamics vs short run dynamics for defense which is not well analyzed in the paper.
Figure 8 in Appendix D shows a plot for increased accuracy as the number of steps increases.
Can short run dynamics also be effective in defending by varying the hyper parameters (step size) of the Langevin dynamics after training EBM?
An in-depth understanding of this effect is missing.

I will be open to changing my score based on the response from the authors.

---

> ### Author Response · Authors · 2020-11-13
> **Response to Reviewer 2**
>
> Thank you for your time and for raising important concerns about our work.
>
> 1. There are two crucial differences between our paper and Srinivasan et al. 2019. The first difference is that Srinivasan et al. only use a few Langevin steps (at most 25) to defend images. We observe that short-run trajectories cannot effectively remove adversarial signals regardless of EBM, as we show in Table 1 and Figure 8. Srinivasan et al. report less than 1% robustness for CIFAR10 against the BPDA+EOT attack. The second difference is that Srinivasan et al. use a Denoising Auto-Encoder to approximate the gradients of an implicitly defined EBM. The Langevin dynamics for this model are not stable enough for long-run defensive sampling. The central contribution of our work is showing that EBM defense can be effective, but only with sufficiently long Langevin trajectories with convergent models.
>
> 2. We do not claim that learning realistic images and stable long-run sampling is impossible with SGD alone (Nijkamp et al. 2019 show that this is possible). Nonetheless, identifying the precise SGD schedule for learning an EBM with both high quality images and stable long-run sampling is difficult, and one typically experiences good short-run image quality at the expense of stability or vice-versa. By dividing the learning process into two distinct stages, one for realism and another for stability, it becomes much easier to balance these factors. Our Adam-SGD learning method shows it is possible to “salvage” non-convergent EBMs and stabilize their long-run samples without affecting the quality of short-run samples.
>
> 3. Thank you for raising concerns about ambiguous use of the terms short-run and long-run. We agree that it would be useful for readers if we define these terms more formally in revisions. Short-run sampling refers to trajectories of approximately 1 to 150 Langevin steps. Short-run trajectories generally do not exhibit noticeable over saturation, but they also do not provide significant defensive benefits. In this work, a long-run trajectory refers to a trajectory that is long enough for defensive properties to emerge, which we find takes several hundred steps (about 1500).
>
> 4. The images used to make the plot in Figure 3 are natural images. The same behavior occurs for adversarial images. Some adversarial images are easily classified with well separated logits (right histogram). Some adversarial images are on the borderline of correct classification (left histogram). The figure is meant to give an idea of easily classified or borderline images, whether they be natural or adversarial, using the EOT defense approximation in Equation 5. We used H=100 rather than H=150 because this diagram was made prior to deciding upon our final defense parameters. The result for H=150 would be very similar. We will make a new version of Figure 3 with H=150 in future revisions.
>
> 5. Our convergent EBM is essentially a convergent version of the unconditional IGEBM. Our work and IGEBM both use standard EBM training methodology introduced by Xie et al. 2016. Besides the Adam-SGD switch, the major learning differences are a) we use a much smaller network to speed up Langevin dynamics by a factor of 30 and b) we do not rejuvenate persistent samples from noise because we observe this often leads to a non-convergent outcome, and c) we do not use an L2 penalty on the energy values because it is unnecessary. We attempted to learn a convergent version of JEM but the results tended to either have stable long-run sampling but low accuracy as a classifier or high accuracy but unstable sampling. Adding the classification loss to the unsupervised ML loss appears to further complicate convergent learning.
>
> 6. We agree that Figure 8 captures that most important aspects of our paper and we will give the figure and discussion in Appendix D more prominence in revisions. The Langevin step size that we use is close to the maximum possible step size. The maximum possible step size depends on the width of the most constrained dimension of the local energy landscape (Neal 2010 https://www.mcmchandbook.net/HandbookChapter5.pdf in Section 5.4.2.2). As observed by Nijkamp et al. 2019, the EBM should be learned using a step size that is close to the threshold where learning becomes unstable. The optimal value of the Langevin step size for EBM learning depends on the geometry of the true but unobserved image density q(x). For CIFAR-10 images with pixels in the range [-1, 1], the optimal step size is close to 0.01. Using a step size significantly greater than 0.01 during training will cause the learning process to fail completely. Likewise, using a step size much greater than 0.01 after the model has been trained will lead to unrealistic images because the resolution of the Langevin dynamics will not be fine enough to accurately follow the energy landscape geometry. Decreasing step size slows the Langevin mixing process and more steps are needed for defense.

---

### Official Review · AnonReviewer3 · 2020-10-28
**A nice paper with impressive performance and comprehensive evaluation**

**Rating:** 9
**Confidence:** 3

**Review:**

This paper proposes an interesting application of Langevin sampling with the energy-based model (EBM) to defend against adversarial attacks. Compared to adversarial training (AT), the proposed adversarial preprocessing (AP) based method can be used to secure existing models without the need for retraining.

As an EOT based defense, naturally, a larger number of replicates would lead to better adversarial robustness. The idea behind the proposed defense is to leverage the long-run sampling using EBM for the EOT based defense. However, training long-run EBM is challenging. To overcome this, the authors apply an interesting two-phase training method with Adam for the first phase to quickly train the short-run one and SGD for the second phase to gradually align the long-run one with the short-run one.

The authors compared their approach to a large number of defenses as well as the state-of-the-art AT based defense. The reported results are very impressive, with an 84.12% benign accuracy and 78.9% adversarial robustness on CIFAR-10. In addition, the proposed method can still achieve 54.9% adversarial robustness even under an adaptive attack, which is close to the state-of-the-art AT based defense. However, compared to AT, the benefit of the proposed method is that it does not require retraining the model.


Strength:
- Achieve robustness performance similar to adversarial training without tampering with the classifier.
- An interesting two-phase training method to address the non-convergent problem for long-run sampling.
- Consider an EOT attack as the adaptive attacking method.

Weakness:
- Overall, it is hard to find flaws in this paper. The only concern I have is that the timing overhead is still large (~10 sec per image when K=1500, Fig. 10) and seems can hardly be parallelized since the long-run sampling is serial. This might greatly limit the applicability of the proposed defense when timing constraint is critical.

---

> ### Author Response · Authors · 2020-11-13
> **Response to Reviewer 3**
>
> Thank you for your positive evaluation of our work. We agree that the most notable limitation of our work is the relatively expensive runtime cost. Attacking a large classifier model with PGD is also an expensive procedure, and we find that we can defend an image using 1500 steps in approximately the same time that it takes to attack the base classifier using 50 PGD steps. Therefore, we can secure images at approximately the same rate that an attacker can generate images against the base classifier. Our defense is significantly faster than the time it takes for an attacker to generate optimal BPDA+EOT adversaries because the sampling dynamics must be incorporated into that attack. Reducing the computational cost of EBM defense is an important direction for future work. We also note that our method might become much more practical with faster compute and mass parallelization that could arise from future technologies. In the present work, our goal is to provide a solid proof-of-concept that EBM defense can secure naturally trained classifiers with a robustness that is competitive with AT.

---

### Official Review · AnonReviewer1 · 2020-11-01
**Interesting paper but needs more study**

**Rating:** 5
**Confidence:** 4

**Review:**

This paper discusses using a classifier + EBM  to construct an expectation of transformation (EOT) defense. The simple overview is that given an input image (potentially corrupted by an adversary), they use long-run Langevin dynamics seeded at the image to generate samples from the data distributions for the corresponding input image and then use samples to estimate EOT.
1) A similar approach was used by Grathwohl et a. (2020) to improve the robustness of their model, so I was wondering if the authors can discuss their novelty here?

2) The paper is not very clear in terms of the methods, descriptions, and explanation of the tables. For example, Section 4.1 refers to "EBM defense" but has not been defined separately, so I assumed it is the defined EOT defense in 3.1 that uses EBMs as T(x).

3) The classification accuracy of Grathwohl et al. (2020) against PGD in Table-1 is not consistent with what is reported in the paper. Grathwohl et al. (2020) reported that JEM-10 is a robust model against PGD attack (Figure-5 in their paper). Would you please discuss these inconsistencies?

4) The authors claim that EOT defense is their contribution, is there any discussion on why you only tried EBMs as T? Why not VAEs? VAEs are also capable of generating samples given the corrupted images, and the sampling process is much faster than long-run Langevin dynamics!  Their output may not be as sharp as EBMs, but does it matter here?

5) It would be nice to use your EBM (without the classifier) as a standalone model in Table-1 to see the comparison with IGEBM.

6) What if you use WideResNet with IGEBM? This actually shows the importance of your long-run sampling.

7) Could you present the effect of H for defense against PGD? I am interested to know how diverse \hat{x}_h are when you run convergent Langevin dynamics seeded at x.

---

> ### Author Response · Authors · 2020-11-13
> **Response to Reviewer 1**
>
> Thank you for your time and for bringing up insightful critiques of our work.
>
> 1. The central novelty of our work is that we show that sufficiently long Langevin trajectories can be used to secure a naturally trained classifier model. Prior works studying defense with an EBM a relatively small number of steps (25 steps in Srinivasan et al 2019., 10 steps Du and Mordatch 2019 and Grathwohl 2020). We find that the Langevin trajectories used in prior defenses are ineffective and that successful defense requires several hundred steps (we use 1500, see Figure 8) until the trajectory can effectively efface adversarial signals from the initial state. Trajectory length is a key ingredient which was not identified in prior works. The stability of Langevin paths is a central problem for EBM learning (Nijkamp et al. 2019). The vast majority of EBMs exhibit over-saturated samples after only a few hundred steps and very few authors examine the distribution of long-run images, the crucial component of our EBM defense. We modify the convergent learning process in Nijkamp et al. 2019 to show that an EBM with highly oversaturated samples can be “frozen” into an EBM with realistic long-run samples without sacrificing sample quality. This observation greatly alleviates the problem of balancing realism and stability of long-run samples, making it much easier to train models that are suitable for defense. With enough Langevin steps and stable sampling paths, EBM defense can be very effective.
>
> 2. The EBM defense refers to the defense in Section 3.1 (specifically equation 4, approximated by equation 5) where T(x) is Langevin sampling for K=1500 steps with the EBM. We will be careful to present this more clearly in revisions. Algorithm 1 describes our attack method and Algorithm 2 describes our training method, and code for both algorithms are provided in the supplementary ZIP file. We will do our best to clarify our methods and presentation of results in revisions. Please let us know if there are other sources of confusion that we should address.
>
> 3. Although our evaluation of the JEM defense is inconsistent with the evaluation in Grathwohl et al. 2020, our results are nearly identical to the results from Croce and Hein 2020 (https://arxiv.org/pdf/2003.01690.pdf in Table 3). While the reason for the inconsistency is unclear, it might be due to the fact that Grathwohl et al. attacked their model by differentiating through the Langevin trajectory. Although this attack is theoretically ideal, it might not be effective in practice due to the complexity of second-order differentiation the Langevin gradient term. The naive PGD attack used by us and by Croce and Hein is much more effective against the JEM model. This is discussed briefly in Appendix G.
>
> 4. VAE’s are known to be susceptible to adversarial attacks that can completely change image class (see Kos et al. 2017 https://arxiv.org/pdf/1702.06832.pdf). Defenses that preprocess with an auto-encoder can be broken by back propagation through the concatenation of the auto-encoder and the classifier (see Athalye and Carlini 2018 https://arxiv.org/pdf/1804.03286.pdf attack on the HGD defense). The trajectory of a Langevin path with an EBM has unique defensive properties. First, the trajectory is memoryless and eventually the sample will not depend on the initial state. Second, the Langevin trajectory exhibits chaotic behavior, so that an adversary cannot reliably climb up the loss landscape filtered through the EBM. These perspectives for Langevin sampling as a defensive tool are discussed in Appendix C.
>
> 5. The IGEBM model used in the defense experiments from Du and Mordatch, 2019 is a class-conditional EBM while our model is an unconditional EBM. Our EBM cannot directly be used as a classifier, unlike the IGEBM defense. Long-run defense with a conditional EBM is problematic because long-run states can quickly change classes if initialized in a separate class than the conditional model class. Our unconditional defense depends on all classes being preserved after a few hundred steps.
>
> 6. We are now performing this experiment with the original IGEBM model and a Wide ResNet classifier and will include it in revised versions. We note that the non-convergent model used in Figure 8 is trained with essentially the same method as the unconditional IGEBM model. Figure 8 shows that a non-convergent model that results in quick oversaturation is not suitable for adversarial defense. Figure 8 further shows that short-run sampling does not lead to effective defense, but sufficiently long-run sampling with a convergent model (about 1500 steps) can provide significant defense.
>
> 7. Thank you for the suggestion, we are now making this diagram. The paths are quite stochastic and using H that is too low will result in many false breaks cause by randomness and not the adversarial signals. This happens when equation 5 is a poor approximation of the true classifier in equation 4.

---

### Official Review · AnonReviewer4 · 2020-11-03
**A new heuristic adversarial defense without convincing empirical results.**

**Rating:** 4
**Confidence:** 2

**Review:**

This paper proposes a defense that uses MCMC sampling with an Energy-Based Model as a preprocessing before classification.
The stochastic transformations used in preprocessing are sampled from the Langevin updates, which has been shown to be useful for adversarial defense in prior work.
Since there is no theoretical guarantees, I consider this method a heuristic approach with some empirical success.
However, there are already many empirical pre-processing defenses, and most of them have been broken by newer attacks.
It is not convincing enough that the proposed defense will be different. For example, one can design better approximations for the transformation operations in BPDA, instead of using the identity function. Actually, when the transformation is larger, the identify function can be a very poor approximation.
Also, the proposed method is not as robust as a variant of adversarial training (Zhang et al. 2019) as shown in Table 2. The authors claim that the main benefit of their method is that it can protect a pre-existing natural model. However, from the run time analysis, it seems that it is even slower than adversarial training, so I am skeptical about the practical value of this method.

---

> ### Author Response · Authors · 2020-11-13
> **Response to Reviewer 4**
>
> Thank you for your time and important questions about our work. We address several points of your evaluation:
>
> 1. Although several prior works use Langevin sampling with an EBM as a defense tool (Srinivasan et al. 2019, Du and Mordatch 2019, Grathwohl et al. 2020), further investigation reveals that Langevin sampling does not provide significant robustness in these works. Srinivasan et. al report that BPDA+EOT breaks their defense on CIFAR-10. The defense from Grathwohl et. al can be broken by a PGD attack against the base classifier, as found in both our experiments and the evaluation by Croce and Hein, 2020 (https://arxiv.org/pdf/2003.01690.pdf in Table 3). Robustness from the defense of Du and Mordatch comes from the base classifier alone, and not from Langevin sampling as we show in Table 1.
>
> 2. We acknowledge that our defense is a heuristic defense. We note that current certified/provable defenses (such as Cohen et al. 2019) are robust with respect to a specific attack (e.g. an L2 attack) and that robustness might not hold for other norm-bounded attacks or more general attacks. Recent theoretical results show that current smoothing methods cannot achieve high provable accuracy for the l_inf norm for data of sufficiently high dimension (see Kumar et al. 2020 in https://arxiv.org/pdf/2002.03239.pdf). Certified defenses under a specific norm are effectively heuristic defenses against attacks that use other (possibly undiscovered) types of imperceptible perturbations. In our view, the question of general adversarial robustness can currently only be evaluated at the level of heuristic defense.
>
> 3. We agree that it is possible that our defense could be broken by a different attack than the BPDA+EOT attack we used. It is true that the BPDA attack might be stronger if a different differentiable approximation besides the identity transformation is used. On the other hand, Grathwohl et al. differentiate through the entire Langevin transformation with K=10 purification steps to evaluate their model but the attack is actually weaker than a naive PGD attack that does not incorporate sampling dynamics. We strongly believe that our adaptive attack represents the current state-of-the-art method for evaluating our defense. We encourage researchers to investigate attack methods that could be used to break our defense.
>
> 4. We acknowledge that the defense runtime is a drawback of our method. Still, our work is a valuable proof-of-concept for a pre-processing defense approach that is distinct from adversarial training and related methods that alter classifier training. We also note that generating PGD samples against a large classifier is a slow process. Our defense runtime with K=1500 Langevin steps takes about the same time as a 50-step PGD attack against the base classifier. This means we can defend an image at approximately the same rate that an attacker can generate an adversarial image using PGD against the base classifier, and at a much faster rate than an attacker can generate BPDA+EOT samples. Further reducing the runtime of the defense is an important direction for future work.

---

### Decision · Program_Chairs · 2021-01-07
**Final Decision**

**Decision:**

Accept (Poster)

**Comment:**

Although the technical novelty is not very high, the finding that long-run Langevin dynamics with convergently learned model provides comparable defense performance to adversarial training will give some impact to the community.